# Compress to Think, Decompress to Speak: Dual-Mode Reasoning in Transformers

## Abstract

Latent reasoning has emerged as an alternative to reasoning with natural language and involves feeding back the last layer's hidden state representation (soft token) to the input of the transformer. This idea is promising, since soft tokens have increased expressive capacity compared to tokens from the vocabulary (hard tokens). Existing works on training transformers with soft tokens often suffer from performance loss and do not allow for sampling of different reasoning traces. We propose a training paradigm for transformers that uses soft tokens, in which the model learns to operate in two modes; one that processes the soft tokens (latent thinking mode) and one that decompresses the soft tokens into few reasoning steps with hard tokens from the vocabulary (local decoding mode). We focus on logical and math reasoning tasks, and fine-tune pretrained models of different size. Our method achieves similar or better performance, compared to supervised fine-tuning with chain-of-thought data across all tasks; while it requires reduced KV cache and allows sampling different reasoning traces at inference. [1]

## 1 Introduction

Reasoning models often rely on increased test-time compute through methods like Chain of Thought (CoT) prompting (Wei et al., 2022); since they have been either trained on or guided by CoT data, which encourages them to generate intermediate reasoning steps in order to reach an answer. Even though CoT data have been shown to improve performance on reasoning tasks, one potential downside is that models are constrained to reason only using tokens of the vocabulary (hard tokens). Projecting the hidden states of a model to the vocabulary, can be thought as a discretization step, which potentially leads to information loss. Reasoning in the latent space has emerged as an alternative, in which the hidden states are not always realized as hard tokens, potentially leading to more expressive and thus shorter reasoning chains. Towards this direction, recent works have sought to

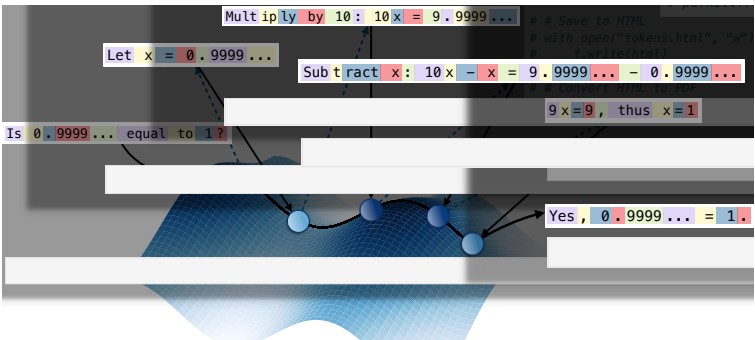

Figure 1: When solving the reasoning task, the model can choose to reason with the hard tokens in the vocabulary space (top), or reason with soft tokens in the latent space (bottom). Our model can traverse either on the latent or in the vocabulary space: processing the soft tokens, or decompressing the soft tokens into the CoT steps. This can be viewed as an interpolation between hard token and soft token methods.

---

[1]Our code is available at this link.

internalize CoT steps by distilling them into the latent space (Deng et al., 2023; Yu et al., 2024; Deng et al., 2024).

One particular approach involves the use of so-called soft tokens, which are not part of the discrete vocabulary of hard tokens. Soft tokens can be constructed in different ways: as weighted linear combinations of the embeddings of the top-$k$ next tokens or of parallel reasoning traces (Zhang et al., 2025; Gozeten et al., 2025); by employing a VQ-VAE trained on CoTs (Su et al., 2025); or by leveraging hidden representations from intermediate layers (Cheng & Durme, 2024). Another line of work uses last-layer hidden states as soft tokens. Hao et al. (2024) took initial steps in this direction, proposing a training algorithm that gradually replaces CoT steps with soft tokens, without explicitly training them. Later, Shen et al. (2025) introduced an auxiliary loss to distill hidden representations from a teacher model trained on explicit CoTs. More recently, Hwang et al. (2025) proposed a three-stage training method: (i) fine-tuning a pretrained model on the target task, (ii) training an encoder–decoder to compress and decompress CoT steps into latent representations, and (iii) training a latent model guided by the learned encoder–decoder. In this work, we focus primarily on the line of work where soft tokens are the last-layer hidden states.

These methods often result in performance loss, compared to explicit CoT training or require training more than one models. More importantly, models that reason only in the latent space produce deterministic trajectories and do not support sampling at inference. This conflicts with post-training algorithms like GRPO (Liu et al., 2024) that require exploring multiple candidate reasoning paths. In theory, soft tokens could learn to encode the joint distribution of all possible reasoning traces, eliminating the need for applying such post-training algorithms. However, because the number of possible reasoning paths grows exponentially with the number of steps, this would require infinite precision for the learned soft tokens. Thus, sampling would still be a desirable feature of a model trained to reason in the latent space. In all of the works described so far, even though sampling could be possible, their performance in such a scenario has not been studied.

Our approach mitigates the concerns raised above by training *one* transformer model without any modification in the architecture that performs in a dual nature; *the latent thinking mode*, in which the model learns to generate the next soft token and *the local decoding mode*, in which the model learns to decode the soft tokens into CoT step(s). The model can also choose to update the soft tokens, using the representation generated in the local decoding mode. Our main contributions are as follows:

- **Dual-mode Operation.** A *single* transformer is trained to operate in both hard and soft token spaces via two modes: *latent-thinking* (predict a soft token given the compressed context) and *local-decoding* (decompress a soft token into a few chain-of-thought steps and update the soft token). If the local-decoding length is zero the model performs latent-only reasoning. Our methods allow training a mixture of the two, by choosing whether or not to update the soft token with some probability. We show promising results that the model can operate in a mixture of the two modes.

- **Sampling through local decoding.** We sample hard tokens in the local decoding step and then update the soft token from these sampled tokens. Thus, stochasticity at the token level induces stochasticity in the soft tokens, yielding latent-level exploration compatible with multi-sample post-training method like GRPO.

- **Reduced KV-cache with no performance loss.** On logical and math reasoning benchmarks, our method matches the greedy decoding and `pass@k` of explicit CoT while reducing KV-cache requirements, enabling more efficient inference.

- **Generalization to harder settings.** Our method has better generalization than CoT-trained models on problems with reasoning chains longer than those seen during training.

## 2 RELATED WORK

**Latent Reasoning via Tokens.** The idea of augmenting LLM's capabilities by providing special tokens in the input, such as pause tokens (Goyal et al., 2024), dot tokens (Pfau et al., 2024), and latent tokens (Sun et al., 2025) has been shown to improve reasoning with minimal training or architectural changes. A related line of work studies soft tokens —continuous embeddings not tied to the discrete vocabulary. Soft tokens are constructed by multiple ways; (i) by superposition over input embeddings (Xiong et al., 2024), or (ii) by using last-layer hidden states (Hao et al.,

2024). Following the superposition approach, recent works show that soft tokens can encode parallel reasoning traces (Gozeten et al., 2025) and can often be used out-of-the-box without any fine-tuniing of pretrained LLMs (Zhang et al., 2025; Wu et al., 2025).

Our work is closer to the second approach, which feeds the model's final-layer hidden states back into the input. This idea was first investigated by (Hao et al., 2024) to compress explicit chain-of-thought (CoT) into a fixed number of soft tokens, and has since been extended to pretraining (Tack et al., 2025) and theoretical analysis of soft-token expressiveness (Zhu et al., 2025a). Concretely,(Hao et al., 2024) adopts stage-wise compression of CoTs into soft tokens, while Shen et al. (2025) introduce an additional distillation loss that further improves performance. A recent work by Hwang et al. (2025) trains an encoder–decoder to compress chain-of-thoughts (CoTs) into soft tokens and, with the decoder frozen, trains a latent-thinking model to generate them using cross-entropy and a contrastive alignment objective. While similar in spirit, our method uses a single end-to-end model that handles both soft and hard tokens. Beyond these, several methods leverage learned soft embeddings from other sources: Cheng & Durme (2024) train models to predict intermediate hidden representations of a frozen LM; Su et al. (2025) use VQ-VAE to discretize and compress CoT embeddings into soft tokens.

**Internalizing Thinking Process.** Beyond using special tokens for latent reasoning, a broader thread seeks to internalize Chain of Thoughts (CoTs) into latent representations. Deng et al. (2023) distill hidden states from explicit reasoning trajectories into standard forward passes, and Deng et al. (2024) extend this by progressively removing explicit CoT while preserving performance. To transfer "System 2" computation into a faster "System 1" mode, Yu et al. (2024); Liao et al. (2025) distill step-by-step reasoning; Su et al. (2024) propose trace dropping for fast-mode inference; and Xia et al. (2025) skip tokens deemed less informative during verbal reasoning. In parallel, looped models (Dehghani et al., 2018; Giannou et al., 2023) iteratively update hidden states without outputting tokens, enabling variable computation depth. This idea has been studied on synthetic tasks for its ability to extrapolate and approximate fixed-point solutions (Schwarzschild et al., 2021; Yang et al., 2024; Gao et al., 2024; Fan et al., 2025), and scaled to large-model pretraining, where the looped model improves the model's reasoning ability (Saunshi et al., 2025; Geiping et al., 2025). These findings are consistent with evidence that greater depth benefits reasoning-related problems (Zhu et al., 2025b; Ye et al., 2024; Sanford et al., 2024).

**Transformers with Memory.** Another line of work relevant to our work is that of augmenting transformers with some form of memory of the past context. The work of Dai et al. (2019) stores the hidden representations of all the layers for past tokens and combines them in the calculation of the attention with those of the new tokens, allowing for information to flow even when the context exceeds the sequence length. Bulatov et al. (2022), introduces recurrent memory transformer (RMT) which use the last-layer hidden representations and appends them in the beginning and end of each segment and iteratively updates them. Later Bulatov et al. (2023) scale RMT to million tokens context. Chevalier et al. (2023) fine-tune pretrained models by using adaptive instead of a constant number of memory tokens and showed benefits for long context length. Our work follows a similar idea in using the soft tokens to summarize the context / CoT steps. However, we focus primarily on reasoning tasks, and demonstrate the effectiveness of soft tokens in compressing the CoT steps with no performance drop. Another line of work, considers training models to replace the CoT steps with verbal summarizations, by learning patterns during training Yang et al. (2025),Yan et al. (2025).

## 3 OUR METHOD

In this section, we describing our training algorithm for reasoning tasks and techniques we employed to train the soft tokens using supervision only on the hard tokens.

**Description for reasoning tasks.** In Figure 2 we provide a depiction of how our method works. In detail, given CoT training data, in which the steps are explicitly defined, we split them (together with the label) into $k$ chunks, where $k$ is the maximum number of soft tokens used. Each chunk is either empty, contains one, or more steps of CoTs. At the end of each chunk we append a special [switch] token. While in the beginning of the answer we have a special token [ans] to distinguish it from the CoTs.

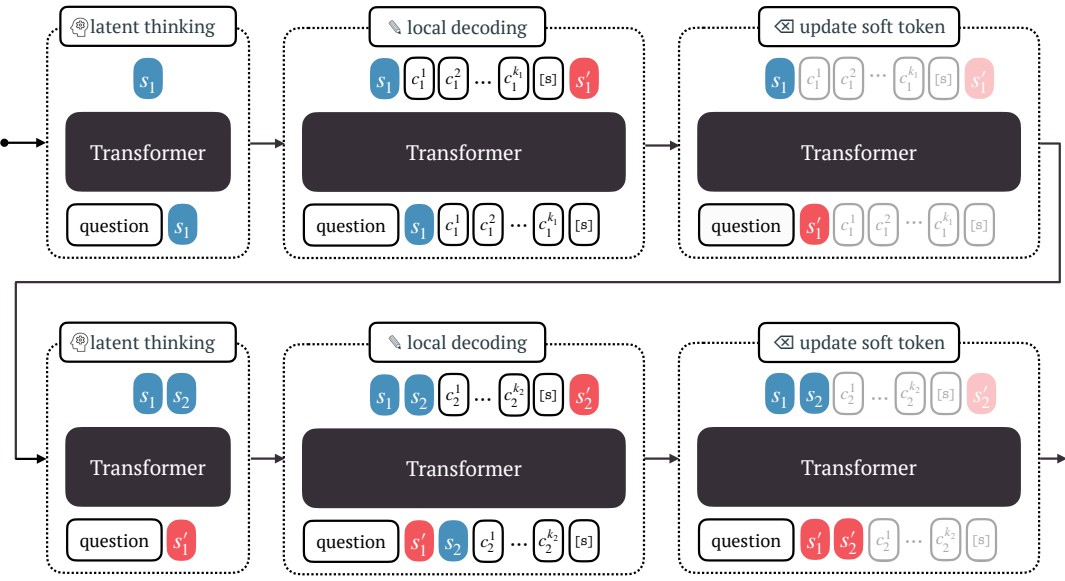

Figure 2: *Training framework of our method.* We train a *single* transformer to operate in two modes. In the *latent thinking* mode, given the question and compressed context, the model predicts a new soft token. In the *local decoding* mode, it decompresses the soft token into a few chain-of-thought steps until a [switch] token is produced, after which the model choose to update the soft token using the representation at [switch].

We start with the latent thinking mode, given a question $q$, the first soft token is produced by passing in the model the question and keeping the last layer's hidden representation ($s_1$) corresponding to the last token. Afterwards, we enter the local decoding mode in which the first chunk of data is fed to the model together with the question and the first soft token. The model predicts some labels which we keep to apply loss. The last layer's hidden representation of the [switch] (noted as $s'_1$ in Figure 2) is used to update the soft token. The steps are repeated until all the chunks have been used. For general reasoning data that do not have explicitly defined steps, we could use a specific delimiter, such as '.' or \n, to separate them into steps.

Notice that this method does not apply any explicit loss on the soft tokens, but these are implicitly trained, since they are needed for the decoding to be successful. We consider the forward pass on the soft tokens as the *latent thinking* mode and the forward pass for decoding as the *local decoding* mode. At inference, the same principle is applied, in which the model switches from the *local decoding* mode back to the *latent thinking* mode, once the [switch] token is predicted. We provide a detailed algorithm of how the method works in Appendix A.2.

**Enhancing soft tokens.** In order to enhance reasoning through soft tokens and learn more robust representations by allowing the model to sometimes skip the local decoding step, we introduce a stochastic soft token updating rule to the above training and inference algorithm. Specifically, during training with probability $p_{\text{update}}$ we update the soft token with the last layer's hidden representation of the [switch] ($s'_1$) and with probability $1 - p_{\text{update}}$ we do not update it (keep $s_1$). At inference the user can pick a new probability $p_{\text{update}}$, and enable or disable the generation of hard tokens to speed up inference. Notice that if probability $p_{\text{update}}$ is 0.0 then the proposed method boils down to only using soft tokens for reasoning. To determine whether the current soft token marks the start of the answer, we always predict a hard token from the soft token. If that token is the start-of-answer marker [ans], we proceed to decode and generate the answer.

**Scheduling on soft tokens.** During training the model needs to learn to handle and adapt in two different tasks; one the use of the soft tokens and two the actual reasoning task. To improve the training stability and learning curve, we apply a curriculum on the number of soft tokens. We start the training with fewer number of soft tokens to keep training closer to supervised fine-tuning with explicit CoTs, in which case one soft token needs to encode multiple CoT steps. As the training

progress we gradually increase the soft tokens used, until we reach the maximum number of soft tokens that we want the model to adapt to. Notice that the max number of soft tokens is chosen such that each soft token compresses at most one CoT step. If the number of soft tokens $k$ exceeds the number of steps, the extra soft tokens are simply not used. The training steps for each stage of the scheduling are decided as a percentage of the total number of steps. In the next section, we have performed an analysis for GSM8k on how the performance changed depending on the percentage used.

**Reduced KV-Cache at Inference.** If the probability of updating the soft token is set to be $p_{\text{update}} \in [0, 1]$, meaning we always generate the hard tokens, then our method for inference uses KV-cache which scales as $q + k + L$, where $q$ is the size of the length of question, $k$ is the number of soft tokens used, $L$ is the maximum length of any generation. The inference FLOPs are scaling as

$$\sum_{l=1}^{q} l + \sum_{i=1}^{k} [(q+i) + p_{\text{update}} \sum_{j=1}^{L} (q+i+j)] = \mathcal{O}(q^2 + k^2 + p_{\text{update}} kL(k+L+q)$$

Similarly, for the standard model that is fine-tuned in the CoTs the KV-cache scales as $q + KL$, while the FLOPs scale as

$$\sum_{l=1}^{q} l + \sum_{i=1}^{kL} (q+i) = \mathcal{O}(q^2 + qkL + k^2 L^2)$$

*Remark* 1. (Relation to memory-augmented transformers) Our approach is closely related to recurrent memory transformers, especially Autocompressor (Chevalier et al., 2023). That line of work primarily extends effective context length by updating learned memory tokens and prepending them to subsequent segments. In our setup, the soft token plays an analogous role: a learned hidden state that compresses intermediate chain-of-thought into a single embedding. We were not aware of this work until recently and our approach came from a different perspective: rather than increasing the retrievable context length, we study how such latent states can compress CoT traces while *preserving accuracy and reducing KV-cache*. Furthermore, the update probability $p_{\text{update}}$ interpolates between memory-augmented processing and purely latent soft-token computation. The success of this approach to different domains further enhances its potential as a general purpose method.

## 4 EXPERIMENTS

We evaluate our method by fine-tuning pretrained language models on mathematical and logical reasoning benchmarks.

### 4.1 EXPERIMENTAL SETUP

We fine-tune GPT2-small (124M) (Radford et al., 2019), Gemma3 (270M) (Team et al., 2025), and Qwen2.5 (0.5B) (Team, 2024) on four datasets: GSM8k (Cobbe et al., 2021), iGSM (Ye et al., 2024), ProsQA (Hao et al., 2024), and ProntoQA (Saparov & He, 2022). For GSM8k we adopt the augmented split of Deng et al. (2024); for ProsQA we follow Hao et al. (2024). For iGSM we use two settings: *medium* with maximum number of operations $op \leq 15$ and out-of-distribution (OOD) evaluation at $op = 20, \cdots, 27$, and *easy* with $op \leq 9$ and OOD at $op = 12, \cdots, 19$. Dataset statistics are reported in Appendix A.1.

Our main baseline is supervised fine-tuning (SFT) on explicit Chain of-Thought traces. To ensure a fair comparison, we use identical hyperparameters for our method and the SFT baseline: we apply learning rate 2e-4 when finetuning GPT2 and 5e-5 when finetuning Gemma3 or Qwen2.5 models. On GSM8k we finetune for 15 epochs for GPT2, and 5 epochs for Gemma3 and Qwen2.5, while for ProsQA and ProntoQA we finetune for 60 epochs. We apply weight decay 0.01, cosine annealing learning rate scheduler, and effective batch size 32.

Regarding the hyperparameter that is specific to our method: Let $p_{\text{update}}$ (Section 3) denote the probability of updating the soft tokens during training; we set $p_{\text{update}} = 0.5$ for ProsQA and $p_{\text{update}} = 1$ elsewhere. Soft token scheduling is enabled by default, where the max number of soft token is set such that for the given task, each soft token encode at most 1 CoT step. We analyze the effects of $p_{\text{update}}$ and the impact of scheduling in Section 4.3.

## 4.2 PERFORMANCE ON REASONING TASKS BY DIFFERENT MODELS

> **Finding 1:** *Across different models and tasks, similar performance to the stronger baseline–either explicit CoT or prior soft-token methods.*

The results on different reasoning tasks and different models are reported in Table 1. On **GSM8k**, for GPT2-small, our method outperforms the CoT baseline and the other soft token approaches. On **ProntoQA**, our approach matches CoT at 100%. On **ProsQA**, latent/soft token methods generally beat CoT, likely because the task benefits from exploring alternatives without committing to a single path. Mirroring this behavior by using $p_{\text{update}} < 1$, our method achieves significantly better performance compared to CoT and competitive with prior soft token approaches.

Across the three backbone models, our method achieves similar or better accuracy to the CoT baseline. The gain is largest for GPT2-small, smaller for Qwen2.5-0.5B, and mixed for Gemma3-270M. We hypothesize that the benefit of soft tokens depends on the vocabulary: GPT2 uses around 50k-token vocabulary, whereas Gemma3 and Qwen2.5 use larger vocabularies (>150k).

| Model | GSM8k | ProsQA | ProntoQA |
|---|---|---|---|
| CoT | 44.73 | 84.6 | **100** |
| iCoT | 30.0 | **98.2** | - |
| Coconut | 34.1 | 97.0 | 99.8 |
| CODI | 42.9 | - | - |
| SbS | 40.3 | 92.6 | - |
| Ours | **47.84** | 94.6 | **100** |

| Model | CoT | Ours |
|---|---|---|
| GPT2-small | 44.73 | **47.84** |
| Gemma3-270M | **48.75** | 47.92 |
| Qwen2.5-0.5B | 58.22 | **58.45** |

Table 1: *(left)* Performance of GPT2-small on GSM8k, ProntoQA and ProsQA with different soft token/latent methods, and *(right)* Test accuracy on GSM8k for GPT2-small, Gemma3-270M and Qwen2.5-0.5B. Note We did not re-implement these methods except for SFT on CoTs; we report the results that each paper reported (iCOT: Deng et al. (2024), Coconut: Hao et al. (2024), CODI: Shen et al. (2025) and SbS Hwang et al. (2025)).

> **Finding 2:** *Better generalization to problems with more reasoning steps than the ones seen during training.*

**Task Difficulty Generalization.** We further evaluate on iGSM (Ye et al., 2024), a GSM8K-style synthetic benchmark that controls the number of arithmetic operations ($op$) required to solve each problem. Unlike Ye et al. (2024) in which the models are trained from scratch, we fine-tune a pre-trained GPT2-small model on iGSM-Easy ($op \leq 9$) and iGSM-Med ($op \leq 15$) datasets, and then test on out-of-distribution (OOD) data with longer reasoning chains (Easy: $op \in \{12, \cdots, 19\}$; Med: $op \in \{20, \cdots, 27\}$). Due to the fact that GPT2-small uses learned absolute positional encodings, length extrapolation is limited: when the CoT spans more steps than seen during training, accuracy drops as expected for both methods. As shown in Fig. 3, our method matches the CoT baseline on in-distribution problems and consistently retains higher accuracy on OOD problems; moreover, its accuracy decays more slowly as $op$ increases. These trends indicate improved generalization on longer/harder reasoning problems when intermediate steps are compressed into soft tokens.

> **Finding 3:** *Test-time performance matches the CoT baseline.*

**Test-time Performance.** As mentioned earlier, our method naturally supports sampling via local decoding and the update of the soft token accordingly. We evaluate `pass@K` and majority voting against the baseline. As shown in Fig. 4, with GPT2-small trained on the GSM8k dataset, our method exhibits the same monotonic increase of accuracy, with increasing $k$ in both `pass@K` and majority voting, indicating it can be applied to post-training on reasoning tasks. Interestingly, even though our method is slightly worse on Gemma3 under greedy decoding (Table 1, right), with temperature sampling it performs well and is more robust to higher temperatures: when $T$ increases from 1.0 to

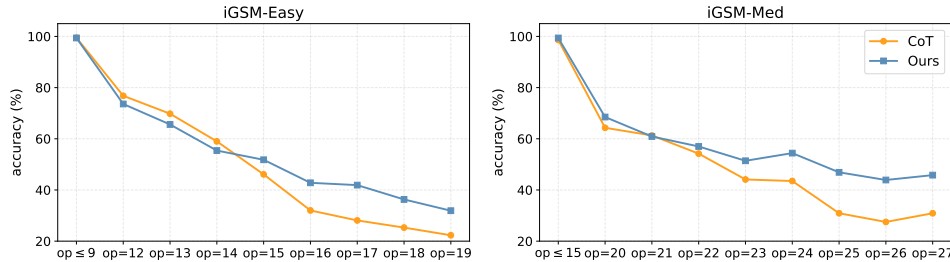

Figure 3: *In-Distribution and OOD Performance of iGSM Easy and Medium Dataset.* Across both settings, our method retains higher accuracy than CoT under OOD shifts with longer reasoning chains and degrades in a slower rate as the number of operations increases.

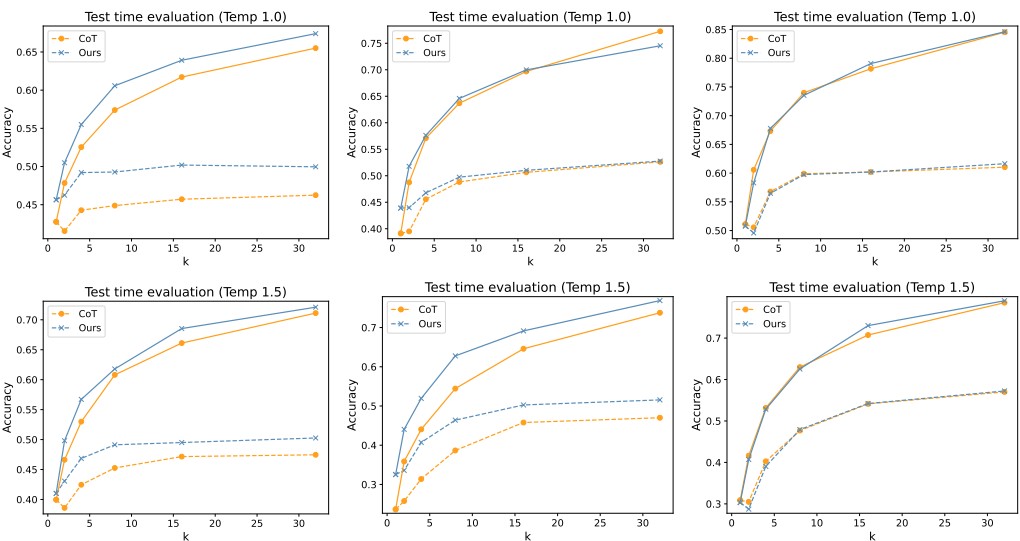

Figure 4: *Pass@K (solid line) and majority-vote (dashed line) results on GSM8K for GPT2-small (left) Gemma3-270m (middle) and Qwen2.5-0.5B (right).* Our method shows monotonic gains as $k$ increases, following the same trend as the CoT baseline. Temperature is set at 1 for the top and 1.5 for the bottom.

1.5, our method maintains similar performance, whereas the CoT baseline drops. More results on different temperature are shown in Appendix A.3.

## 4.3 EFFECTS OF HYPER-PARAMETER CHOICES

Our method makes the following changes to standard SFT on explicit CoT: (i) we apply soft token scheduling with a maximum budget $k$, and (ii) we optionally update each soft token during training with probability $p_{update}$. In this section, we study the effects of these hyperparameters on GSM8k and ProsQA.

**Effects of number of soft tokens.** In GSM8k, we set the max number of soft tokens to be $k_{max} = 12$, since most GSM8k CoT traces contain $\leq 10$ steps, this allows (when possible) for at most one soft token per step. To study the effects of a smaller soft token budget, we also evaluate $k_{max} \in \{4, 6\}$, where some soft tokens must summarize more than one step on average. We report the performance of different numbers of soft tokens in Table 2. As shown in the table, encoding fewer steps per token improves reasoning performance, and one-per-step suffices in matching and surpassing the performance with CoT.

**Effects of scheduling.** We train end-to-end to jointly learning the target task and generation of soft tokens. Using fewer soft tokens keeps training closer to the explicit-CoT baseline, but restricts

| | CoT | 4 soft tokens | 6 soft tokens | 12 soft tokens |
|---|---|---|---|---|
| GPT2-GSM8k | 44.73 | 42.68 | 44.2 | **47.84** |

Table 2: *Performance of GPT2-small on GSM8k for different number of maximum soft tokens with scheduling and probability $p_{update} = 1.0$.*

the amount of time in which the model is using the maximum number of soft tokens. We therefore study how different schedules during training, affect the performance of the model. Specifically, we increase $k$ by one every $N\%$ of total steps until reaching $k_{\max}$; for the remaining steps of training we use $k_{\max}$. As shown in Table 3 , an interval of about 6% gives the best GSM8k accuracy, balancing task learning and soft token learning; smaller or larger intervals still lead to performance close (or even better) to the explicit CoT baseline. Following this rule, we use a scheduling interval of $\approx 6\%$ for both Qwen2.5 and Gemma3.

| Percentage of total steps | 0.0% | 4.4% | 5.6% | 6.7% | 8.3% |
|---|---|---|---|---|---|
| GPT2-GSM8k | 43.44 | 43.9 | **47.84** | 46.32 | 47.38 |

Table 3: *Performance of GPT2-small on GSM8k for various scheduling schemes with max soft token budget $k_{\max} = 12$. The percentage of the steps corresponds to the amount of steps performed for increasing the number of soft tokens by one. The remaining steps are performed with keeping the number of soft tokens to be the maximum. The first column (0.0%) corresponds to using the maximum number of soft tokens from the beginning.*

> **Finding 4:** *Operating in both modes can help increase performance in tasks that soft tokens outperform the baseline.*

**Effect of the soft token update probability.** During training, we stochastically update the soft tokens via local decoding with probability $p_{update}$. We study the effect of $p_{update}$ on ProsQA, where soft token method outperform the explicit CoT baseline. We sweep $p_{update} \in \{0, 0.2, 0.5, 0.7, 1\}$, where $p_{update} = 1$ corresponds to always use local decoding and update the soft tokens. As shown in Table 4, training and evaluating with $p_{update} = 0.5$ yields the best performance while reducing the expected number of hard-token decoding steps by 50% on average.

| Probability of updating | 0.0% | 20% | 50% | 70% | 100% |
|---|---|---|---|---|---|
| Always decoding | 0.0 | 91.0 | 91.6 | 87.4 | 82.0 |
| Probabilistic decoding | 83.0 | 91.6 | **94.6** | 89.6 | 72.0 |
| Latent only | 90.6 | 89.0 | 94.0 | 93.4 | 0.0 |

Table 4: *Performance on ProsQA of GPT2-small with variable soft token update probability $p_{update}$ (columns) and across test-time strategies (rows). At test time, "Always decoding" always performs local decoding; "Probabilistic decoding" performs local decoding with probability $p_{update}$ matching the column; and "Latent only" uses only soft token updates. For training with probability 1 and 0, probabilistic decoding is reported with probability 0.5. The best result is **94.6%** at $p_{update} = 0.5$ with probabilistic decoding, which halves the expected number of hard-token decoding steps.*

## 5 LIMITATIONS AND DISCUSSION

**Training efficiency.** Our method uses the soft tokens in an auto-regressive loop, so each subsequent soft token is a function of the previous soft tokens, this leads to a higher memory requirement (empirically observed to be less than 2x). This limitation can be easily mitigated by using back propagation through time with stopping gradients (BPTT) (Sutskever, 2013), a technique that in generally is employed for recurrent architectures. Furthermore, the forward pass during training requires $k$ forward passes, since the input is segmented into $k$ parts, thus training time scales roughly

linearly with the average number of soft-tokens used. We mitigate this with a curriculum on the maximum number of soft tokens, yet a single-pass baseline still remains faster. This motivates future work on approximating soft-token updates to enable parallelization and improve training efficiency. Regarding the stochastic update rule, we believe it can be replaced by a learned, input-dependent $p_{\text{update}}$; in that case the model would operate in a dual mode and would decide on its own when to exit one mode and enter another one.

**Scaling and post-training.** In this work we study models up to 0.5B parameters. We plan to scale this work to billion-parameter models and larger reasoning benchmarks, and to integrate RL-based post-training (e.g., GRPO) to probe how latent thinking interacts with reinforcement learning. At larger scales we will also examine stability across longer latent unrolls, including variants of BPTT. For these scales, parameter-efficient fine-tuning such as LoRA (Hu et al., 2022) would allow for faster training, while also we may consider lightweight adapters specialized for the latent-thinking and local-decoding modes within a single model.

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

## A  APPENDIX

### A.1  DATASET DETAILS

For GSM8K, ProntoQA, and ProsQA, we adopt the preprocessing and train/validation/test splits of Hao et al. (2024). For iGSM, we generate the dataset using the official open-source implementation [2]. We record the data split in Table 5, and the max number of CoT steps in each dataset in Table 6. For the OOD dataset in iGSM, the number of CoT step is the number of the operation $op$, so the iGSM with $op = 20$ contains 20 CoT steps.

Table 5: Dataset splits.

| Dataset | Training | Validation | Test | OOD |
|---|---|---|---|---|
| GSM8k | 385,620 | 500 | 1,319 | - |
| ProntoQA | 9,000 | 200 | 800 | - |
| ProsQA | 17,886 | 300 | 500 | - |
| iGSM-med/easy | 1,498,500 | 500 | 1,000 | 1,000 |

Table 6: Statistics of Dataset.

| Dataset | Training | Validation | Test | Example CoT step |
|---|---|---|---|---|
| GSM8k | 13 | 8 | 8 | `<<12+3=15>>` |
| ProntoQA | 11 | 11 | 11 | `Each yumpus is a wumpus.` |
| ProsQA | 6 | 6 | 6 | `Every hilpus is a numpus.` |
| iGSM-med | 15 | 13 | 13 | `Define Niagara Falls Aviary's Enclosure as y; so y = b = 20.` |
| iGSM-easy | 9 | 9 | 9 | `Define Goat Cheese's Rye as S; so S = 3.` |

### A.2  TRAINING DETAILS

Here we present the detailed training and inference algorithm in Algorithm 1 and 2.

---

**Algorithm 1** Training

1: **Input:** data point $(q, y)$, Transformer $TF_\theta$, probability $p_{\text{update}}$.
2: Choose $k$ {number of soft tokens}
3: $\{x_j\}_{j=1}^{k} \leftarrow$ RANDOMPARTITION$(y, k)$ {split $y$ (CoTs + labels) into $k$ parts}
4: $h_0 \leftarrow TF_\theta(q)[-1]$
5: **for** $j = 1, \dots, k$ **do**
6:    $z_j \leftarrow TF_\theta([q, h_{0:j-1}, x_j, \texttt{[switch]}])$
7:    **if** $\mathcal{U}(0, 1) < p_{\text{update}}$ **then**
8:       $h_{j-1} \leftarrow z_j[-1]$ {update soft token}
9:    **end if**
10:   $t_j \leftarrow$ PROJ$_{\text{vocab}}(z_j[:-1])$
11:   $h_j \leftarrow TF_\theta([q, h_{0:j-1}])[-1]$
12: **end for**
13: $\mathcal{L} \leftarrow$ LOSS$(\{t_j\}_{j=1}^{k}, y)$

---

**Algorithm 2** Inference

1: **Input:** $q, TF_\theta, p_{\text{update}}, max_k$.
2: $i \leftarrow 0, h_0 \leftarrow TF_\theta(q)[-1]$
3: **while** $i < max_k$ and $t \neq \texttt{[eos]}$ **do**
4:   $t \leftarrow$ PROJ$_{\text{vocab}}(TF_\theta([q, h_{0:i-1}])[-1])$
5:   **if** $\mathcal{U}(0, 1) < p_{\text{update}}$ and $t \neq \texttt{[ans]}$ **then**
6:     T $\leftarrow []$
7:     **while** $t \neq \texttt{[switch]}$ **do**
8:       T.append$(t)$
9:       $z \leftarrow TF_\theta([q, h_{0:i}, \text{T}])[-1]$
10:      $t = $ PROJ$_{\text{vocab}}(z)$
11:     **end while**
12:    $h_{i-1} \leftarrow z[-1]$ {update soft token}
13:   **end if**
14:   $h_i \leftarrow TF_\theta([q, h_{0:i-1}])[-1]$
15:   $i \leftarrow i + 1$
16: **end while**

---

[2] https://github.com/facebookresearch/iGSM

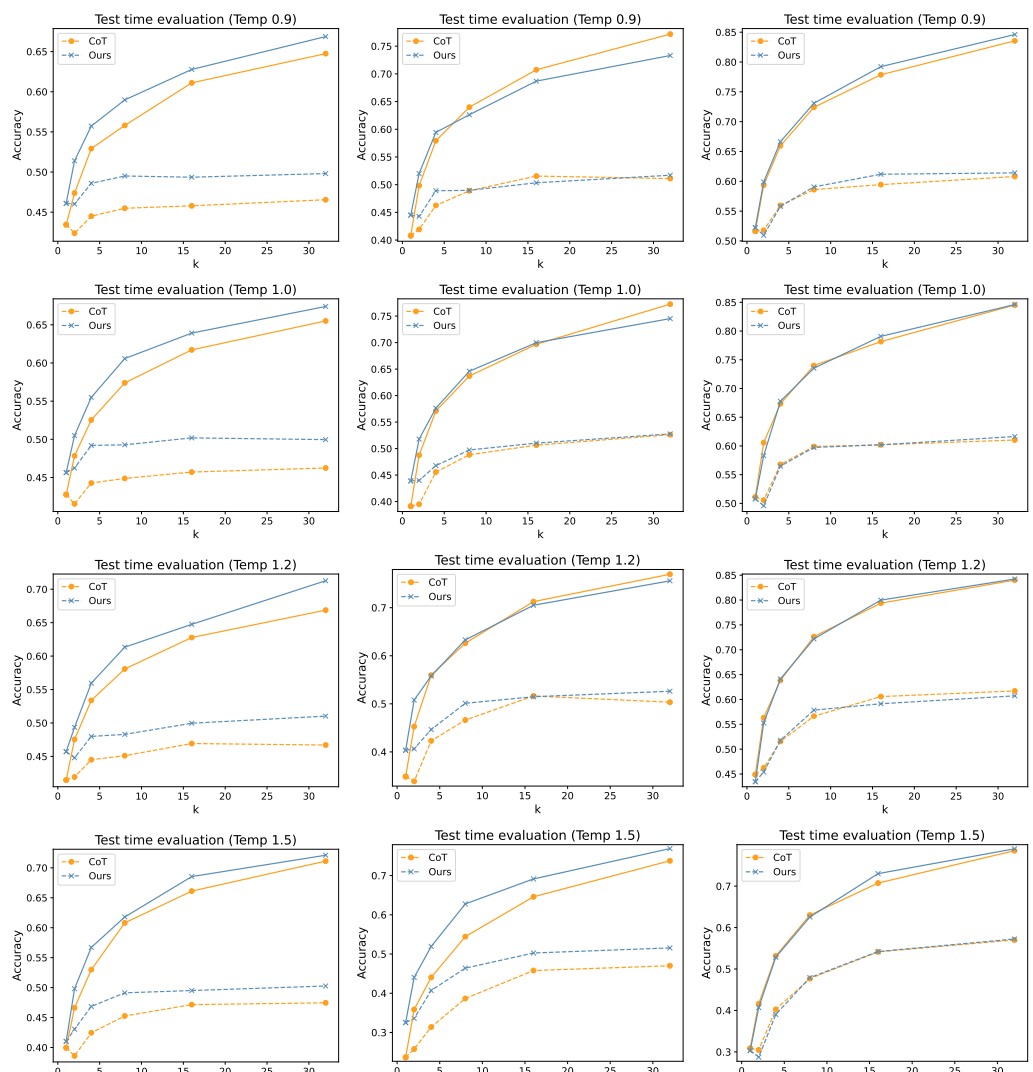

Figure 5: Additional results for GPT2 small, Gemma3-270m and Qwen2.5-0.5B on majority voting and `pass@K`, using different temperatures. Both methods follow a similar curve.

### A.3 ADDITIONAL EXPERIMENTAL RESULTS

**Test-time performance.** In Section 4.2, we study the test time evaluation performance of our method and the baseline CoT on `pass@k` and majority vote. Here we present the full results on varying temperature, completing the results demonstrated in the main text. As shown in figure 5, our method exhibits similar trend in both metrics. Interestingly, on Gemma3 model, even though our method is slightly worse in greedy decoding, as the temperature increases, we observe an improved performance on both `pass@K` and majority voting.

**Soft token update probability for GSM8k.** In the main text we examined $p_{\text{update}}$ on ProsQA, where skipping local decoding (i.e., performing a latent update instead) helps the model learn useful soft token representations. Here we extend the analysis to GSM8k. We train with $p_{\text{update}} = 0.5$ and vary $p_{\text{update}}$ at inference. As shown in Table 7, GSM8k accuracy drops whenever $p_{\text{update}} > 0$, indicating that inserting latent-only steps at test time hurts performance relative to always decoding. We hypothesize this effect is task-dependent: tasks like ProsQA benefit from encoding parallel reasoning traces in the latent space, so using $p_{\text{update}}$ during training and inference can help; in con-

trast, GSM8k appears to benefit from grounding intermediate steps in hard tokens, so nonzero $p_{\text{update}}$ reduces accuracy, however with the trade-off of generating fewer hard tokens.

| Probability of updating | 0.0% | 20% | 50% | 70% | 100% |
|---|---|---|---|---|---|
| GPT2-GSM8k | 25.24 | 31.31 | 37.91 | 40.79 | 44.807 |

Table 7: *Performance of GPT2-small on GSM8k for variable soft token update probability.*

