# OpenReview forum: "Compress to Think, Decompress to Speak: Dual-Mode Reasoning in Transformers"
_ICLR.cc/2026/Conference — ICLR 2026 Conference Withdrawn Submission_

### Official Review · Reviewer_LfrN · 2025-10-20

**Soundness:** 1
**Presentation:** 1
**Contribution:** 2
**Rating:** 2
**Confidence:** 4

**Summary:**

This paper attempts to mix both soft and hard tokens in chains of thought using a Coconut style approach for their soft tokens.
The method works by generating a step and then producing a [switch] token, which then overwrites the prior generated step with a soft token.
Overwriting is stochastic during training so the model learns to use this at test time.

The paper uses three models (all <500M parameters) and most results focus on GPT2-127M.
The authors show their method is superior for GSM8K for GPT2 and Qwen and comparable on ProsQA and ProntoQA.
Using the GPT2 model the authors then evaluate out of distribution and using Pass@K and majority voting.
The authors then ablate the performance of GPT2 over different numbers of soft token and varying how the soft token is inserted during training.

**Strengths:**

- Interesting approach to increase reasoning within language models
- Performs well for GPT2 models
- Good positioning within the related works and future work.

**Weaknesses:**

- The paper only studies small models, and the authors state using LoRA to scale to larger models is future work. I feel larger models are needed in this version of the paper to confirm the results scale. Moreover, most experiments are conducted with GPT-2 which is not a state of the art base model.
- The paper is limited to GSM8K evaluations in almost all cases.
- Efficiency measures are missing in all cases. For example, of the results in Table 1 how many FLOPs were required for each?
- There is no legend or indication as to what the letters mean in Figure 2.
- It would be nice to provide similar equations to the ones on lines 228 and 234 for regular transformers for the reader to quickly compare.
- Fixing the hyperparameters for both CoT and soft CoT methods (line 260), seems skewed towards one approach or the other. Which method was the hyperparameter search conducted on?
- I am not too sure what I am supposed to get out of the 2 rows in Figure 4. This plot could do with having shared y-axes within the columns and grid lines for easier comparison across plots for model types.
    - My main issue here is it is that the text says that the temp=1.5 line for Gemma (middle) is better for the new method than at temp=1.0 but the values in the temp=1.5 plot seems to be worse than in the temp=1.0 plot. Hence, I disagree with the conclusions drawn from this plot when presented in its current form. Updating and emphasising the values in this plot so we can see the (significant, beyond error bounds) improvement would resolve this. An additional 3 column, 1 row plot with all 8 lines on it would also be good for the appendix.

**Questions:**

1. Is your evaluation set up identical to all the methods presented in Table 1? If there are slight evaluation changes this could largely impact the reported accuracies, I would suggest rerunning these evaluations for the baselines instead of copying them.
2. Is the KV-cache impacted when the soft tokens are decoded into hard tokens during local decoding?
3. Training for 5/15 epochs on gsm8k is a lot. Why is this required?
4. On line 158, it states the steps are explicitly defined in the training data, does this mean using grammar to separate data into steps?
5. Why are ProsQA and ProntoQA not considered for Gemma and Qwen in Table 1?
6. What are the error bounds on Table 1, for example is the 0.23% increase for Qwen significant?
7. In Table 2, 12 soft tokens to be best. Is this because it uses more compute?
8. Table 3, 5.6% and 8.3% are very close at 47.84 and 47.34, is this significant? Why should we choose 5.6 over 8.3? Same for table 4, "latent only" achieves 94% only 0.6% lower than "probabilistic".

---

### Official Review · Reviewer_zgjF · 2025-10-25

**Soundness:** 2
**Presentation:** 2
**Contribution:** 3
**Rating:** 2
**Confidence:** 4

**Summary:**

The paper proposes a method for training transformers that mixes soft tokens (latent representations) and hard tokens (vocabulary tokens). The model learns two modes: one for latent reasoning using soft tokens, and one for local decoding that turns these into a few reasoning steps in natural language. The main goal is to match or outperform CoT fine-tuning while being faster and more memory efficient (smaller KV cache) and allowing sampling of different reasoning traces.

**Strengths:**

* Overall results show a performance increase over SFT baselines
* Ideas overall are nice; having the mix of hard and soft tokens increases the representational power of the input (in theory)
* Generalization to harder settings.
* Seems to outperform baselines on **GSM8K**, **ProQA**, and **ProntoQA**.

**Weaknesses:**

* Writing
  * Figure 2 does **not** explain what ( \c_{i} ) is.
  * Figure 1 does not add value to the paper and is possibly not entirely correct.
* This method should, in theory, be inherently capped by the CoT data. I am concerned that the current SFT baseline is weak. Comparing to GRPO would be a more meaningful comparison.
  * Are there any experiments that show latent reasoning is doing something different than the CoT distillation?
* Model size: the model size seems quite low, which makes me suspicious that this will generalize to larger models.

**Questions:**

The *p-update* of 0.0 is soft token, does that mean that *p = 1.0* allows hard tokens? I found the writing around this part confusing, as it’s unclear where the *p-update* is coming from. Is *p-update* compared to the probability of the switch token? Line 267–269 says *p-update* is 1.0 — does that mean only hard tokens are generated? I would assume not. However, these clarifications greatly detract from the value of the paper.

---

### Official Review · Reviewer_Kx5F · 2025-10-31

**Soundness:** 2
**Presentation:** 1
**Contribution:** 2
**Rating:** 2
**Confidence:** 4

**Summary:**

This work proposes a technique for combining discrete and latent CoT reasoning to improve the flexibility and efficiency of reasoning LLMs. They propose a training scheme that learns to use both latent computation and discrete computation from existing CoT traces, without requiring any ground truth supervision for the latent reasoning states. They evaluate their proposed approach on GPT, Gemma, and Qwen models of up to 500M params and demonstrate parity with standard discrete CoT in some cases and advantages over existing latent CoT methods in most cases.

**Strengths:**

1. The proposed training method achieves the new latent reasoning ability despite the lack of ground truth labels required to do so.
2. The approach appears competitive with the selected baselines on the selected evaluations.

**Weaknesses:**

### Clarity issues with the methodology description

### 1.

Sec 3 prose and diagram in Fig 2 are confusing. For instance in the first panel, why is s_1 shown twice, isn't it generated for the first time in this first step with only the question as input? then in the next panel (top middle) why are the c_i's visually left shifted at the top, is this meaningful? Normally, there is also a right shift in target token indices which is what I think is meant to be shown to accompany L190 " The model predicts some labels which we keep to apply loss." but it's not clear exactly what this loss is and what goes into it at each step and what is being compared to what to compute the loss.

I took a look at the algorithm in A.2 but it doesnt match the notation in Fig 2. Since you have space in the draft, can you promote the Algo to main body Sec3, and make sure that it is notated using the same variables and indexing as Fig 2, and try and refine the prose description from L157 to L200?

### 2.

Related question, despite what L195 says, it is unclear how error propagates to the previous latent s'_1 when locally decoding chunk c_2. As L195 states having an accurate s'_1 is "needed for decoding to be successful" but errors in decoding c_2, caused by inaccuracies in s'_1, don't seem to have a way to propagate back to the forward pass that created s'_1 because the intermediates c_1 are no longer in the context window during the loss calculations for the c_2 decoding stage. Perhaps this can all be clarified with the above incorporation of the formal Algo block, and a unification of notation.

Note at the _very_ end of the draft, back propagation through time is mentioned, which suggests a missing piece of the puzzle above ... During training, is the computational graph maintained that chains the forward pass that produced s'_1 into the forward pass being used to decode block c_2 and then produce s'_2, and then that composed graph again tied to the forward producing s'_3, etc. so that when backward is finally called, all of the error assignment to the computation producing s'_2, and s'_1, via intermediates c_2 and c_1 is correctly backpropagated?

### Lack of empirical quantification of any efficiency benefits over standard CoT or other prior methods

### 3.

Can test time efficiency be reported clearly somewhere? L224 asymptotic argument is not sufficient on its own. It is unclear how to compare the proposed method and the text CoT in the real evaluation settings. For Table 1, no comparison is discussed at all (k is not even mentioned), but in Fig 4, since the x-axis is shared, then these values are in some way meant to be comparable, but it is not clear how this should be interpreted. Perhaps the inference flops and total KV-cache size could be plotted as alternate x-axes for Fig 4? This would clearly show whether or not the proposed method is more compute or memory optimal than the standard CoT. Because the orange and blue lines are quite close shifts left or right in one curve or the other could make the difference between the proposed technique and the baseline more or less favorable.

### 4.

For clarity, train time costs of the standard CoT and and the proposed method need to be reported. Gesturing at truncated backprop through time as a "easy mitigation" to the inefficiencies is not sufficient to assuage any worries about cost because the method could immediately fail to work if this technique is used based on the reviewer's current understanding of the composed computation graph and how this is the only way for the s'_i's to be properly learned.

### 5.
Baselines other than text CoT are not reproduced, just copied from prior work, so whether or not the comparisons are quite fair against those prior latent CoT methods is unclear

**Questions:**

1. L218, should this say "Notice that the max number of soft tokens is chosen such that each soft token compresses at ~most~ _least_ one CoT step."?

2. L 267 why is 0.5 used for ProsQA but 1 is used for all other evaluations?

---

### Official Review · Reviewer_sJq4 · 2025-10-31

**Soundness:** 1
**Presentation:** 1
**Contribution:** 1
**Rating:** 2
**Confidence:** 4

**Summary:**

This paper proposes a framework to train transformers to operate in two modes: latent-thinking where the model predicts a soft token, and local decoding where the model decodes the soft tokens into text tokens. The authors fine-tune models up to 0.5B parameters (GPT2, Gemma, Qwen) on math and logical reasoning tasks. They claim their method achieves similar or better performance than standard CoT supervised fine-tuning (SFT) while reducing KV cache requirements at inference, enabling sampling of diverse reasoning traces, and, most notably, improving out-of-distribution generalization to problems with longer reasoning chains.

**Strengths:**

The paper is well motivated and well suited in the latent reasoning literature. The core idea of letting one model switch between latent reasoning and decoding into text tokens flexibly is relatively novel. Empirical evidences suggest that the method performs on par with the baselines in terms of task performance while being more generalizable to longer reasoning steps at test time.

**Weaknesses:**

1. As shown in Table 7, it seems like the benefits of latent reasoning is highly task dependent. For example, the best performance on GSM8k is without latent reasoning at all. It's therefore unclear how generally applicable the proposed methods is. Such major limitations should be more explicitly discussed in the main text.
2. The training and inference algorithms are not clearly explained in the main text. Although the algorithms are provided as part of the appendix, I find them a bit hard to read (not all the notations are clearly defined).
3. The experiments are conducted on small models up to 0.5B. It's unclear how this method can be stably scaled to larger models.
4. The paper lacks a principled justification for its convoluted dual-pass training. Why is this specific process (Pass 1 for loss, Pass 2 for next state, stochastic update) necessary? Some ablation studies would be very helpful.

**Questions:**

1. Could you provide some analysis on the computational costs of the training process?
2. Could you provide more details about the training and inference algorithms in the main text?
3. One of the core benefit of the method is its generalization on number of operations. Could you provide some insights on why it can generalize better?

---

### Note · Authors · 2025-12-01

I have read and agree with the venue's withdrawal policy on behalf of myself and my co-authors.